# Sustainability of an evidence-based intervention supporting transition to independent care for youth living with HIV in Kenya

**Alina Metje**[1]*, **Sarah Shaw**[1], **Cyrus Mugo**[2], **Mercy Awuor**[2], **Annabell Dollah**[2], **Hellen Moraa**[2], **Christine Kundu**[2], **Dalton Wamalwa**[3], **Grace John-Stewart**[1,4,5,6], **Kristin Beima-Sofie**[1☯], **Irene Njuguna**[1,2☯]

**1** Department of Global Health, University of Washington, Seattle, Washington, United States of America, **2** Kenyatta National Hospital, Nairobi, Kenya, **3** Department of Pediatrics and Child Health, University of Nairobi, Nairobi, Kenya, **4** Department of Epidemiology, University of Washington, Seattle, Washington, United States of America, **5** Department of Medicine, University of Washington, Seattle, Washington, United States of America, **6** Department of Pediatrics, University of Washington, Seattle, Washington, United States of America

☯ These authors contributed equally to this work.

* alinametje@gmail.com

## Abstract

Integrating and sustaining evidence-based interventions (EBIs) in routine care is crucial to improving HIV treatment outcomes among youth living with HIV (YLH). However, EBIs are often not sustained post clinical trial. An Adolescent Transition Package (ATP) delivered by health care workers (HCWs) and tested in Kenya in 2021 significantly improved YLH readiness to transition to independent care. Post-trial, participating clinics could continue using the ATP. We conducted 30 in-depth interviews with health care workers to evaluate determinants of continued ATP implementation one-year post-trial. Interviews used semi-structured guides, informed by the Consolidated Framework for Implementation Research version 2.0 (CFIR v2.0). Transcripts were analyzed thematically to identify key influences of ATP sustainment and fidelity post-trial. Effective training during and after the trial, and continued internal and external support for implementation, were crucial for sustained acceptability and feasibility. In contrast, staff shortages and high turnover, lack of integration into the existing electronic medical system, and maintaining staff motivation were barriers to ATP sustainment. Implementation fidelity was limited by workforce constraints and HCW beliefs about the importance of individualizing content and delivery to be responsive to individual client needs. ATP adaptability afforded optimization of delivery to overcome workforce constraints and meet client needs, increasing HCW perceptions of feasibility and motivating continued use. Alignment between observed impact and care provision goals further motivated ongoing ATP utilization. Strategies to ensure continued training and integration of tools into existing systems have the potential to further enhance ATP sustainability.

**Data availability statement:** Excerpts of the transcripts relevant to the study are presented within the paper. All data are available upon request.

**Funding:** This work was funded by a 2021 Adolescent HIV Implementation Science Alliance Award (AHISA) award G-202012-67159 to IN and KBS administered by CRDF Global. This publication was supported in part by Fogarty International Center (FIC) K43TW011422 to IN. Additional support was provided by the UW Global Center for Integrated Health of Women, Adolescents and Children (Global WACh) and the UW/Fred Hutch Center for AIDS Research, an NIH funded program under award number AI027757 to KBS. The funders had no role in study design, data collection and analysis, decision to publish, or preparation of the manuscript.

**Competing interests:** The authors have declared that no competing interests exist.

## Introduction

Despite significant advances toward achieving UNAIDS 95-95-95 goals [1], whereby 95% of persons living with HIV should know their HIV status, 95% of persons diagnosed with HIV should receive antiretroviral therapy (ART), and 95% of all those receiving ART should have suppressed viral loads, youth living with HIV (YLH) remain at substantial risk of adverse outcomes. High HIV incidence coupled with poor clinical outcomes among YLH necessitates an urgent need for effective and sustainable evidence-based interventions (EBIs) [2]. Given the double burden of disease, health care worker shortages and limited resources, sustainment of health interventions in low resource settings, including Kenya, remains particularly challenging [3]. However, there is little evidence on EBI sustainment or determinants of sustainment post-trial despite expected changes in funding, implementation support and implementer motivation [4].

The transition from pediatric to adult care presents unique challenges for YLH. It involves changes in clinics, providers, and provider-client interactions, and is often accompanied by anxiety and reluctance to transition, treatment interruptions, and high mortality [5–7]. A standardized transition plan is a crucial element of successful transition to adult or independent care management [8]. Recognizing the need for structured support during this transition, the Adolescent Transition to Adult Care for adolescents living with HIV (ATTACH) study was conducted. This hybrid 1 implementation-effectiveness cluster randomized clinical trial (cRCT), which evaluated both the intervention effectiveness and factors influencing implementation simultaneously [9], developed and tested an Adolescent Transition Package (ATP) to support Kenyan YLH with transition from pediatric/adolescent focused care to adult care [10]. The ATP was delivered by healthcare workers (HCWs) during routine clinic visits, to YLH individually or in groups. Study results demonstrated significantly higher overall transition readiness among YLH in intervention compared to control facilities [11]. HCWs found the intervention to be acceptable and feasible to implement. Post-trial, intervention sites could continue to deliver the ATP without support from the study team. After demonstrating effectiveness during the RCT, our team disseminated study results to the national HIV program, and the ATP has now been included in revised adolescent HIV guidelines. We anticipate collaborating with the national program to scale the intervention.

Among ATTACH study intervention sites, we evaluated determinants of sustainment (defined as continued program delivery after initial implementation [12]) and fidelity (defined as the extent to which delivery of the intervention adhered to the protocol [13]) of ATP implementation one-year post-trial. Sustainment and fidelity are crucial for ensuring that interventions remain effective and achieve intended outcomes. Using a qualitative approach to evaluate these outcomes provides a deeper understanding of HCW's perspectives and experiences with ATP implementation, which can inform approaches to facilitate its scale-up and long-term integration into routine care in Kenya. Few studies in resource-limited settings address transition, making this program a potential model for other countries to replicate. Lessons learned from ATP delivery may also help inform scale-up and sustainability of other evidence-based interventions in Kenya and similar settings.

## Methods

### Study description

We conducted a qualitative evaluation involving semi-structured individual interviews with HCWs one year after the original ATTACH trial ended. This qualitative evaluation is part of a larger mixed-methods study (ATTACH-Sustain) to identify post-trial determinants

of ATP sustainability and adoption and develop a plan for future intervention scale-up. ATTACH-Sustain is a follow-on study to the primary ATTACH cRCT [11]. The ATTACH study evaluated ATP implementation and effectiveness among 20 clinics (10 intervention, 10 control sites) located in four high HIV-burden counties in Kenya: Homa Bay, Kajiado, Nairobi, and Nakuru. Details of the study procedures and intervention are provided elsewhere [10]. Briefly, the ATP consisted of a chapter booklet and tracking tools delivered by existing HCWs to support YLH in gaining knowledge and skills needed for independent care. After the trial, control sites were provided with the tools and trained to implement them. ATTACH-Sustain was conducted one-year post-trial at all 20 sites and included surveys, qualitative interviews, and a collaborative stakeholder workshop to understand contextual factors affecting sustainability and future scale-up. This manuscript presents results from the qualitative analysis of interviews with HCWs from ATTACH study intervention clinics.

The Consolidated Framework for Implementation Research (CFIR 2.0) [14,15] guided all aspects of this qualitative study, including data collection, data analysis, and reporting of findings. The CFIR 2.0 consists of five domains that organize and influence implementation: innovation, outer setting, inner setting, individuals, and implementation process, and has been recently updated to better reflect low- and middle-income settings, which ensures that the unique challenges and contexts of our setting are adequately addressed.

Research ethics approval for ATTACH-Sustain was provided by the KNH/UoN Ethics and Review Committee, approval number P238/03/2022. Participating HCWs provided written informed consent. County leadership and facility in-charges were engaged in the prior ATTACH trial and provided approval for these follow-on activities.

**Inclusivity in global research.** Additional information regarding the ethical, cultural, and scientific considerations specific to inclusivity in global research is included in the Supporting Information (S1 Checklist).

## Data collection

Using purposive sampling, we recruited two to three HCWs from each of the 10 ATTACH study intervention clinics to participate in semi-structured individual interviews (IDIs). Typically these clinics have a small number of staff, between 3–12 providers, who deliver a full range of services. Sampling of two to three HCWs per clinic allowed us to capture diverse perspectives across multiple clinics and ensure information power [16]. Prior to IDI recruitment, all HCWs from intervention sites completed surveys to capture data on their experiences with ATP delivery (ex: How long have you been using the adolescent transition package tools), professional roles within the clinic (ex: peer mentor, counselor, clinician), and overall satisfaction with the ATP tools (ex: Are you satisfied with the way the Adolescent Transition Package is designed to be delivered). Survey responses were used to recruit a diverse sample of HCWs representing a range of cadres, experiences, and satisfaction scores. Recruitment began September 19th 2022 and concluded February 3rd 2023. Semi-structured IDIs were conducted at a convenient time for the HCWs by one of four trained interviewers (MA, AD, HM, CK). Interview guides were informed by all domains of the CFIR 2.0, and focused on exploring HCW experiences delivering the ATP and the factors supporting continued implementation post trial. Interviewers were all Kenyan women with substantial experience conducting qualitative interviews related to adolescent HIV and grounded in CFIR, previous experience working on the ATTACH study, and who are fluent in languages most commonly used in these settings. Prior to conducting the interviews, the senior qualitative researcher provided training on and reviewed the interview guides with the interviewers to refine the interview tools and ensure comprehension.

Interviews were a median of 31 minutes (range 16–66 minutes). They were audio recorded and conducted in a combination of English and Kiswahili or Dholuo, depending on interviewee preference. All interviews were transcribed verbatim and translated as needed by the qualitative interviewers as they were most familiar with the conversation and context. Structured debrief reports were written by interviewers shortly after IDI completion to capture immediate feedback on interview content and quality.

### Data analysis

We conducted a directed content analysis to identify key determinants of ATP sustainment and fidelity [17]. ATLAS.ti 23, a qualitative data analysis software providing advanced tools for coding, visualizing, and interpreting qualitative data, was used for data management and analysis. A team of three female researchers conducted coding and analysis of the interview transcripts (AM, IN, KBS). The codebook was developed using a hybrid of deductive and inductive approaches. Deductively, predetermined codes based on the CFIR 2.0 were used to develop an initial codebook. The codebook incorporated a majority of CFIR constructs; however, several constructs within the process domain were found to be irrelevant for this analysis and were removed from the codebook prior to coding. Emergent concepts not included in the CFIR-guided codebook were identified through open coding, discussed with the team in real time, and added to the codebook as deemed appropriate [18]. When new codes were deemed necessary, previous transcripts were reviewed to apply the new codes. These inductive codes were added primarily to capture recommendations provided by participants regarding future intervention design and implementation. To ensure coding strategies were consistent and dependable, the team held regular meetings to discuss codes, establish a common understanding, and resolve code interpretation discrepancies. Throughout the coding process, the team continuously reviewed and revised codes and code definitions to ensure accuracy and improve understanding. All transcripts were reviewed multiple times for code consistency, and early coded transcripts were reviewed again at the end of the coding process to account for any changes in code definition interpretations throughout the coding process.

Members of the coding team engaged in ongoing reflexivity, including memoing and group discussions, throughout the entire project to consider their personal biases and discuss alternative interpretations of the data. Post coding, queries by CFIR construct and implementation outcome were used to systematically organize and categorize the data into meaningful concepts to identify themes related to sustainment and fidelity.

## Results

A total of 30 HCWs participated in the study. HCWs were a median age of 34 years, and the majority (73.3%) were female (Table 1). They reported a median of 7 years working with YLH and 2.8 years implementing the ATP. Overall, 77% of the participants had been trained to deliver the ATP by ATTACH study staff, while remaining HCWs were trained by other HCWs in clinics. HCWs represented a range of cadres, including peer mentors (43%), counselors (20%), and clinical officers (13%).

Overall, HCWs shared positive feedback on the ATP and post-trial implementation. HCWs described the ATP as a well- designed and valuable tool for staff and clients, which positively influenced their perceptions of acceptability, feasibility, and appropriateness. HCWs believed ATP tools were easy to use, and had a positive impact on YLH health, all of which facilitated continued implementation post-trial. When evaluating key influences on sustainment and fidelity, we identified several determinants that facilitated or hindered continued implementation (Table 2).

**Table 1. Demographic characteristics of healthcare workers who participated in interviews.**

| Demographic variable | Median (IQR) or N (%) |
|---|---|
| Age | 34 (10.75) |
| Years working with YLH | 7 (3.6) |
| Years implementing ATP | 2.8 (1) |
| Gender | |
| Female | 22 (73.3) |
| Male | 8 (26.7) |
| Trained on ATP delivery by | |
| ATTACH study staff | 23 (77) |
| HCWs | 7 (23) |
| Cadre | |
| Clinical officer | 4 (13) |
| Counselor | 6 (20) |
| Peer mentor[a] | 13 (43) |
| Nurse | 2 (7) |
| Psychologist[b] | 5 (17) |

[a]Peer mentor includes mentor mothers/fathers, operation triple zero (OTZ) champions, peer counselors, and youth champions.

[b]Psychologist also includes psychologists with dual training in counseling or social work.

## Effective training and continued internal and external support are crucial for ensuring long-term acceptability

During the ATTACH study, HCWs at intervention facilities received initial training from study staff followed by six months of supported implementation optimization through continuous quality improvement cycles (CQI). Reflecting on ATP training, almost all participants reported positive experiences, finding trainings informative, comprehensive, and engaging. Training helped overcome initial negative perceptions of ATP tools being complex and burdensome.

> *"The fact that we understood the tools...If we had not understood them, we might have stopped using them. Our knowledge of the tools has made us continue using them."*
>
> *- Counselor, 3 years of ATP experience*

Due to high turnover rates among HCWs, not all HCWs could undergo training by ATTACH study staff. Those who received training from ATTACH staff felt equipped to effectively train new staff and were champions for continued implementation. However, some HCWs not trained during the ATTACH trial described less enthusiasm for ATP tools.

> *"The other challenge is the new staff. As compared to us who were trained by the ATP trainer, they don't seem to understand and embrace the tool as much as we did."*
>
> *- Psychologist and counselor, 3 years of ATP experience*

In addition to initial training, HCWs appreciated CQI meetings conducted during the trial period and the continued insight provided by ATTACH study staff throughout the study period. These positive, supportive interactions helped maintain motivation and provide encouragement to HCWs during the trial and also facilitated sustained use post-trial.

**Table 2. Key factors influencing continued implementation and supporting quotes.**

| Theme | Quote |
|---|---|
| Effective training and continued internal and external support are crucial for ensuring long-term acceptability | "So, at first it felt complicated but after the training, I realized that the tools were very important."<br>- Psychologist, 3 months of ATP experience |
| | "When [study staff] taught me, I understood the tool so I also wanted to pass on the information. That is what helped us keep the momentum."<br>- Youth champion, 7 months of ATP experience |
| | "You see, the ATTACH study was being done in the facility but only a few healthcare individuals were involved, not all of them. So, you see, it is doable but now, it will need either you do an on-the-job training or you do more training for the healthcare workers who are coming in so that they embrace it."<br>- Peer counselor, 2 years of ATP experience |
| Alignment between observed impact and care provision goals as well as intervention adaptability were motivators for continued ATP use | "Once you get used to something, you continue to use it if it had some good impacts."<br>- Clinical officer, 1 year of ATP experience |
| | "We must have a way of sustainability, because at the back of our minds we know that the implementing partner will eventually leave……If it is something that is helpful, we have to find a way of keep on implementing it even after they have left because we still have the adolescents and the transition process must take place……….So, we just took it on our own as something that was helpful to us together with our adolescents."<br>- Peer counselor, 3 years of ATP experience |
| | "That one made it easier when to create the support group for the adolescents, since we had something to talk about. Sometimes we do support groups for different groups… so, it gives the medics ideas on what to discuss with the adolescents and youths."<br>- Clinical officer, 1 year of ATP experience |
| Staff shortages and high turnover, lack of integration into the existing electronic medical system, and maintaining staff motivation present barriers to sustainment | "The challenge that I can talk about was…like you know, it required quite a number of healthcare workers to implement this. But then some of us left along the way."<br>- Counselor, 3 years of ATP experience |
| | "We moved to paperless, but we don't have the ATTACH forms in the [electronic medical] system so I would say that as a challenge."<br>- Nurse, 2 years of ATP experience |
| | "You see when I am around, I can follow up. But when I am not around, you find there are missed opportunities maybe because of the overwhelming group of clients."<br>- Peer counselor, 2 years and 9 months of ATP experience |
| Recommendations for scale-up | "They will give me more attention if we had one in their language. That's the challenge I came across. There are those who may understand better if they can relate more to the tool in their language."<br>- Peer mentor, 5 months of ATP experience |
| | "If it is something that is now being integrated and being rolled out, we must have support from the county level. If it is something that is just happening within the facility and no policies [are] in place at the sub county level to support it, others may feel that it is not their program. But if it is something that has been adopted as a policy trailing from the head to us, then people would see it as something that is helpful."<br>- Peer counselor, 3 years of ATP experience |

*"I can say the support was there. First the training. Second was that small motivation because as you take time to listen or to give views, you're motivated."*

*- Psychologist and counselor, 3 years of ATP experience*

Within the clinic, internal champions, clinic leadership, and teamwork were important in facilitating successful integration of the ATP into workflows. External support from HIV implementing partners or the Ministry of Health (MoH) also played a crucial role in ensuring sustained utilization of the ATP. The level of external support and involvement described by HCWs varied by clinic, ranging from granting permission to implement ATP tools to actively

encouraging their ongoing use. However, some participants highlighted the challenges of coordinating with multiple external agents, mainly when collaborating with partners who prioritize different objectives.

> *"So sometimes, when we look at that like it is for the client, then we work soberly. But the moment we each have to push the organizational objectives, then we begin to have logger-heads. One will want this, one the other one, and the 'grass' who is the client gets hurt, yeah."*
>
> *- Nurse, 2 years of ATP experience*

## Alignment between observed impact and care provision goals as well as intervention adaptability were motivators for continued ATP use

HCWs noted improved clinical outcomes following ATP implementation, including decreased anxiety regarding viral load monitoring, improved viral load suppression, increased willingness to attend clinic appointments and improved self-acceptance among YLH. HCWs also believed that the ATP improved client-provider relationships. Observing the positive impact the ATP had on YLH experiences and clinical outcomes motivated HCWs to sustain use.

> *"And in fact, what I am really happy about too is now they are really open, they are ready to share their concerns, which was not there before. Before you realize an adolescent could come very quiet, an introvert, not wanting to speak, but nowadays, she is the one to tell you that, 'Sister you are not talking about it today. Why? I have one, two, and three we need to discuss about this.'"*
>
> *- Nurse, 2 years of ATP experience*

The benefits HCWs observed among current clients also motivated them to use the ATP with new clients needing disclosure or transition support.

> *"….everyday we get new clients who have not been disclosed and those that need transition. So, there is no way it can be finished, unless we close down the clinic."*
>
> *- Psychologist, 3 months of ATP experience*

HCWs described how these positive outcomes among YLH aligned with clinic goals and objectives, which included a strong commitment to providing client-centered care and prioritizing youth needs and well-being. Clinic goals of providing the best care possible for youth helped HCWs remain open to new ideas and trying new interventions. HCWs also appreciated the positive feedback from YLH and their caregivers on the tools, further motivating continued use.

> *"There is a lot of positive feedback from the clients and even the caregivers. We used to struggle with the caregivers with why they should do the disclosure. After the introduction of the tools, we have a very good guideline. The caregivers have also embraced the tools. The adolescents are so eager about the tools when they come, they even remind you where we had reached with the tools."*
>
> *- Psychologist and counselor, 3 years of ATP experience*

HCWs described how easy it was to adapt the ATP to different delivery formats, which allowed them to manage workforce constraints and meet YLH's needs and preferences. Several

HCWs described adjusting to workload constraints by using the tools in support groups and peer-to-peer education and varying how much content was delivered at a time.

*"Now it is about the patient need, being that I know my adolescents. I know the gaps that each and every adolescent has, so there are some adolescents that force me now to go per chapter, and there are others that I look at the need and sort out."*

*- Nurse, 2 years of ATP experience*

Some HCWs extended use of the tools to adult clients, finding that the educational material included in the ATP also helped when counseling adult clients who were not suppressed, had knowledge gaps, or were newly diagnosed with HIV.

*"And adults need to be taken through that book because there are some adults who do not know why they are taking their medicine."*

*- Counselor, 3 years of ATP experience*

### Staff shortages and high turnover, lack of integration into the existing electronic medical system, and maintaining staff motivation present barriers to sustainment

One common challenge affecting sustainment was the increased workload associated with providing comprehensive adolescent care, particularly in clinics with staff shortages. A few participants highlighted missed opportunities resulting from insufficient staff resources.

*"Yes, it is a bit difficult. Sometimes people get burn out until they don't get enough time to go through the tools."*

*- Clinical officer, 1 month of ATP experience*

Staff turnover was a recurrent challenge, and when new staff members were not adequately trained, they were less inclined to embrace the tools. In addition to challenges of having enough staff, some clinics did not have enough ATP tools, which negatively influenced implementation.

*"…the manuals are not enough, yeah they are few like now we have only two, they have disappeared, and the disclosure we have one which is not enough for us because we are three people implementing yeah. Because when you have 100 clients and only one or two books, then it will not work."*

*- Psychologist, 2 years of ATP experience*

One adaptation that clinics made that was not fidelity consistent was discontinuing use of tracking tools due to limited time and resources available for printing. Tracking tools were paper-based forms, while client management systems were largely electronic. This disconnect in client documentation systems created challenges in consistent completion. One participant mentioned adapting using tracking sheets solely for clients who were lost to follow-up or had a high viral load.

*"So we don't use the files…frequently. You only find us using the files when we are in a very close follow-up. Maybe a lost follow-up client or a brought back to care or unsuppressed.*

*So we don't use. We normally have Kenya EMR [electronic medical records], and EMR has everything."*

*- Peer counselor, 2 years and 9 months of ATP experience*

A few participants noted that the absence of additional financial resources supplied during the clinical trial, such as reimbursement for time spent in meetings discussing ATP optimization and making phone call reminders for YLH to attend visits, made it difficult to sustain motivation for implementing the tools post-trial.

## Recommendations for scale-up

Most participants thought the ATP should be scaled up across all facilities in Kenya, regardless of HIV prevalence, to support HCWs to deliver transition services even when they relocate to different clinics. Alongside expansion throughout Kenya, HCWs emphasized the need to include more language translations to meet needs of diverse populations. HCWs also noted the importance of expanding the availability of the tools, including to other departments, and creating digital versions of the tool to increase availability.

*"We can have it in an online package, a soft copy. For someone who is learning, I think everybody has a phone…I find the booklet being tiresome to carry. So, if I have the package in a phone, in a soft copy or in something, it is easier to let them go through it."*

*- Counselor, 3 years of ATP experience*

Some HCWs suggested integrating the ATP into existing point-of-care systems, such as EMRs, and into MoH policies and engaging MoH in scale-up to ensure its widespread adoption and familiarity amongst clinic staff.

*"So if it's able to go around nationwide, then it's better if we can involve MoH and people in the offices like the government of Kenya… Until they see something like that, they will not see the essence of using it. So it's also better to also involve the government."*

*- Peer counselor, 2 years and 9 months of ATP experience*

## Discussion

Our study identifies several key themes regarding ATP sustainment and fidelity one-year post-trial. HCWs found the ATP to be easily optimized within their setting, facilitating alignment with clinic goals and high acceptability and feasibility one-year post-trial. Initial training by the study team and additional training of new staff supported continued implementation. Supportive internal and external infrastructure and relationships and observed positive outcomes among YLH and their caregivers helped maintain motivation and long-lasting enthusiasm for ATP tool use. However, sustained implementation was negatively impacted by workforce shortages, high staff turnover, and lack of integration into existing EMR systems. Changing and integrating new interventions into the national EMR system requires multiple steps and increased stakeholder engagement, which was beyond the scope of this study. HCWs highlighted that the high adaptability of the ATP allowed them to make changes which would address some of these challenges, thereby facilitating continued use despite constraints.

The ATP addressed gaps in transitional care and significantly improved YLH's readiness to transition to independent care [11]. Ensuring the continuity of evidence-based interventions (EBIs) such as the ATP, beyond clinical trials is crucial for bridging the know-do gap and

improving HIV treatment outcomes for YLH. However, there is evidence that most interventions are not sustained post clinical trial and, therefore, do not translate to directly benefiting clients. Studies to evaluate why interventions are not sustained are often lacking [4,19]. Our study provides evidence of key determinants of sustainment that could translate to other studies evaluating EBIs for YLH in similar settings. We found that effective training helped overcome negative perceptions of the tools and equipped HCWs with the skills and confidence to train new staff. Training is identified as a key component of initial implementation and sustainment of EBIs [20–22]. However, training plans must accommodate existing staff turnover challenges by providing ongoing training and support for new staff and incorporate specific EBI training to sustain implementation [23].

As with other studies evaluating factors influencing sustainment of EBIs [24–26], we found that supportive internal structures helped maintain motivation and long-lasting enthusiasm for ATP tool use among HCWs. The use of internal champions, who are defined as individuals who continually advocate for use of interventions, has proved effective in implementation of prevention of vertical transmission of HIV programs as well as viral suppression among youth [27,28]. In HIV clinics, peer champions could serve this role and have effectively been supportive in delivery of other youth interventions [29,30]. In our study, peers played a crucial role as implementers, acting as champions for the ATP tool. Their active involvement likely played a significant part in sustaining the ATP. Identifying local champions for continued training to preparing them to support implementation efforts may be an effective strategy for sustaining implementation.

Workforce constraints, as we found in our study, are common challenges in health systems in many countries in sub-Saharan Africa [31–33]. Approaches to address this have previously included task shifting, task sharing, and aligning EBI delivery to fit within the existing workforce [34–36]. Examples of successful models include the incorporation of peer support to provide non-clinical educational or support services in clinics [37]. We found that intervention adaptability, including the ability to adapt tasks to fit within workforce infrastructures, was a key driver of sustainment. Clinical trials often have rigid protocols and require additional staff who are not available post-trial. The ATTACH study was a pragmatic trial built to ensure continued use of the tools post-trial, relying almost exclusively on the clinic workforce. Our findings resonate with the broader context of addressing workforce constraints where strategies such as aligning evidence-based intervention (EBI) delivery with existing human resources are essential. Overall, studies on sustainment of EBIs show the importance of training, internal champions, and addressing workforce constraints in maintaining long-term use of interventions. Similar to other studies, our research emphasizes the role of supportive internal structures and champions in sustaining EBIs. Additionally, our study highlights the critical need for adaptable intervention models that align with existing workforce constraints.

The intervention design process is key for facilitating this alignment; a co-designed intervention is likely to fit clinic flow and incorporate views of a wide range of users, making it adaptable to local settings [38–40]. The successful sustainment of the ATP is likely related to the wide range of stakeholders, including the Kenya MoH, who were included in the initial intervention development [41]. As demonstrated by the PrEP roll-out in Kenya [42], government and clinic buy-in, integration of tools within existing national frameworks, and intervention alignment to clinic goals are critical for sustainment [4].

## Strengths and limitations

Findings from this study synthesize experiences of a subset of HCWs from the original ATTACH study intervention sites located in four counties in Kenya whose experiences might not be generalizable to all HIV clinics in Kenya. However, use of the updated CFIR (version

2.0) allows for a standardized assessment of varied implementation contexts and facilitates comparison with other settings, including LMIC settings. Evaluations to understand continued intervention use post-trial are not commonly conducted. The level of sustainment across the health facilities varied and our qualitative study did not capture the extent of sustainment beyond identifying whether individuals were still implementing the tools. However, the in-depth approach to evaluation of sustainment allowed us to characterize what HCWs identified as important contextual factors to them that influenced their perception of sustainment. It's important to note that our assessment of sustainability is limited to one year of implementation at this stage. Further studies are necessary to assess long-term sustainability in delivery beyond this initial timeframe.

## Conclusions

Our findings highlight key determinants of post-trial EBI sustainment, emphasizing the importance of training, internal and external support, and intervention adaptability. These factors should be considered for future intervention development. Strategies to ensure continued training and improve intervention integration into existing point-of-care systems can contribute to enhanced sustainability and scalability. Translating the ATP materials to more languages, creating digital versions, and integrating ATP tools into official policies at the MoH level as part of scale-up efforts could maximize acceptability and further extend ATP impact and reach among YLH. These findings have broader implications for improving long-term HIV care for YLH in sub-Saharan Africa and may be relevant to similar settings.

## Supporting information

**S1 Checklist.  Inclusivity in global research.**
(DOCX)

**S1 File.  Interview guide.**
(DOCX)

## Acknowledgments

We would like to acknowledge all study participants for taking time to contribute to this research. We also thank the study staff at each facility for their dedication to this work. We gratefully acknowledge the Ministry of Health representatives from Homa Bay, Kajiado, Nairobi, and Nakuru counties.

## Author contributions

**Conceptualization:** Dalton Wamalwa, Grace John-Stewart, Kristin Beima-Sofie, Irene Njuguna.

**Data curation:** Sarah Shaw.

**Formal analysis:** Alina Metje, Kristin Beima-Sofie, Irene Njuguna.

**Funding acquisition:** Grace John-Stewart, Kristin Beima-Sofie, Irene Njuguna.

**Investigation:** Sarah Shaw, Cyrus Mugo, Mercy Awuor, Annabell Dollah, Hellen Moraa, Christine Kundu, Kristin Beima-Sofie, Irene Njuguna.

**Methodology:** Cyrus Mugo, Kristin Beima-Sofie, Irene Njuguna.

**Project administration:** Sarah Shaw.

**Resources:** Mercy Awuor, Annabell Dollah, Hellen Moraa, Christine Kundu.

**Supervision:** Sarah Shaw, Kristin Beima-Sofie, Irene Njuguna.

**Visualization:** Alina Metje.

**Writing – original draft:** Alina Metje, Kristin Beima-Sofie, Irene Njuguna.

**Writing – review & editing:** Alina Metje, Sarah Shaw, Cyrus Mugo, Mercy Awuor, Annabell Dollah, Hellen Moraa, Christine Kundu, Dalton Wamalwa, Grace John-Stewart, Kristin Beima-Sofie, Irene Njuguna.

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
