## [Decision Letter · Decision Letter 0]

10 Jun 2024

PGPH-D-24-00904

Sustainability of an evidence-based intervention supporting transition to independent care for youth living with HIV in Kenya

Dear Dr. Metje,

Thank you for submitting your manuscript to PLOS Global Public Health. After careful consideration, we feel that it has merit but does not fully meet PLOS Global Public Health’s publication criteria as it currently stands. Therefore, we invite you to submit a revised version of the manuscript that addresses the points raised during the review process.

Please submit your revised manuscript by Jul 25 2024 11:59PM. If you need more time than this to complete your revisions, please reply to this message or contact the journal office at globalpubhealth@plos.org. Please include the following items when submitting your revised manuscript:

A rebuttal letter that responds to each point raised by the editor and reviewer(s). You should upload this letter as a separate file labeled 'Response to Reviewers'.A marked-up copy of your manuscript highlighting changes made to the original version. You should upload this as a separate file labeled 'Revised Manuscript with Track Changes'.An unmarked version of your revised paper without tracked changes. You should upload this as a separate file labeled 'Manuscript'.

We look forward to receiving your revised manuscript.

Kind regards,

Siyan Yi, MD, MHSc, PhD

Academic Editor

Journal Requirements:

1. Please include a complete copy of PLOS’ questionnaire on inclusivity in global research in your revised manuscript. Our policy for research in this area aims to improve transparency in the reporting of research performed outside of researchers’ own country or community. The policy applies to researchers who have travelled to a different country to conduct research, research with Indigenous populations or their lands, and research on cultural artefacts. The questionnaire can also be requested at the journal’s discretion for any other submissions, even if these conditions are not met.  Please find more information on the policy and a link to download a blank copy of the questionnaire here: https://journals.plos.org/globalpublichealth/s/best-practices-in-research-reporting . Please upload a completed version of your questionnaire as Supporting Information when you resubmit your manuscript.

Additional Editor Comments (if provided):

Reviewers' comments:

Reviewer's Responses to Questions

**Comments to the Author**

1. Does this manuscript meet PLOS Global Public Health’s publication criteria ? Is the manuscript technically sound, and do the data support the conclusions? The manuscript must describe methodologically and ethically rigorous research with conclusions that are appropriately drawn based on the data presented.

Reviewer #1: Yes

Reviewer #2: Yes

Reviewer #3: Yes

Reviewer #4: Yes

Reviewer #5: Yes

2. Has the statistical analysis been performed appropriately and rigorously?

Reviewer #1: Yes

Reviewer #2: N/A

Reviewer #3: Yes

Reviewer #4: Yes

Reviewer #5: Yes

3. Have the authors made all data underlying the findings in their manuscript fully available (please refer to the Data Availability Statement at the start of the manuscript PDF file)?

Reviewer #1: Yes

Reviewer #2: Yes

Reviewer #3: Yes

Reviewer #4: Yes

Reviewer #5: Yes

4. Is the manuscript presented in an intelligible fashion and written in standard English?

Reviewer #1: Yes

Reviewer #2: Yes

Reviewer #3: No

Reviewer #4: Yes

Reviewer #5: Yes

5. Review Comments to the Author

Reviewer #1: The area of study is of high importance in contemporary public health program interventions. The study was subjected to rigorous qualitative methods adopting the Consolidated Framework for Implementation Research. Data analysis (content analysis) using ATLASti was scientifically sound and results were profound and generalizable.

Reviewer #2: Review Reports

Title: Sustainability of an evidence-based intervention supporting transition to independent care for youth living with HIV in Kenya

Review Comments

A. On the Scope: Is that transition of Adolescent youth HIV services to independent care or sustainability of transition of Adolescent youth HIV services to independent care? When do we conduct qualitative study and how did you taken this issue in to account while doing this research? Adolescence is the age of transition hence the specific selection of adolescents with sensitive illness like HIV and the decision capacity is questionable? Why don’t you conduct among other segment of the population E.g., adults? What is the difference and similarity of independent care with self-management practice? Why sustainability or continuity after pot trial? Is that post trial evaluation or sustainability assessment? How do you assess sustainability in the absence of the beneficiaries meaning the adolescents?

B. On the Background and methods

• The background should establish strong rationale for the ‘sustainability’ ‘transition’ of the care and the appropriateness of the framework and its advantage over others?

• The specific type of the study design is not mentioned

• The use of software for analysis is not described.

• Data is not collected from the adolescents and other segments of the population.

• Trustworthiness of the data is lacked

C. On the result and the discussion section

Not presented in logical and clear manner

The tables are not self-explanatory

Needs concise presentation

The discussion is relatively better but still needs more explanation with appropriate argument and refinement.

Regards,

Reviewer #3: language has major issues it needs to be revised

purpose and objectives needs revision

methodology needs more clarity and precision in writing it should separately discuss the setting, design, eligibility and then methods of collection of data

Reviewer #4: Introduction section

Briefly explaining the UNAIDS 95-95-95 goals would provide all readers with context.

Reordering the introduction to start with the specific challenges faced by Young People Living with HIV (YLH) could immediately engage the readers by addressing the main focus of the study. This could be followed by a discussion on the ATTACH study and its outcomes, and then the focus on sustainment and fidelity, creating a clearer narrative.

Explicitly state what makes this study novel or unique. For example, mention if this is one of the first studies to evaluate post-trial sustainment and fidelity of an EBI for YLH in Kenya or similar contexts.

Method section.

Using the Consolidated Framework for Implementation Research (CFIR 2.0) is appropriate and well-justified. However, a brief explanation of why CFIR 2.0 was chosen over other frameworks could enhance understanding

It might also be beneficial to mention any specific training the interviewer received for this study to emphasize their preparedness further.

The median interview duration of 31 minutes is noted, which provides a sense of the depth of the interviews. Including a range (e.g., minimum and maximum duration) could offer additional context on the variability of the interview lengths.

Include examples of specific questions from the interview guide to illustrate how the interviews explored HCW experiences and implementation factors.

The flexibility in conducting interviews in English, Kiswahili, or Dholuo based on interviewee preference is commendable. It ensures that participants can express themselves comfortably and accurately. It would be helpful to elaborate on the translation process to ensure the accuracy and consistency of the transcriptions.

Describe any quality control measures for transcription and translation to ensure data integrity

The coding process is well-described, but it would be helpful to elaborate on how the team handled emergent themes that did not fit within the initial codebook.

Discussion Section

While workforce shortages and high staff turnover are mentioned, discuss specific strategies to mitigate these issues. For example, elaborate on how task shifting and peer champions can be systematically implemented.

The lack of integration into existing EMR systems has a negative impact on sustained implementation. However, the discussion could be strengthened by providing more details on why this integration was challenging and what steps could be taken to address this issue.

The discussion references other studies evaluating factors influencing the sustainment of EBIs. It would be beneficial to provide a more detailed comparative analysis, highlighting how the findings of this study align with or differ from those of other studies.

Emphasize the impact of the findings on future research and practice. For instance, how can the identified determinants of sustainment be applied to other EBIs?

While the discussion focuses on sustainment one year post-trial, it would be valuable to consider the ATP's long-term sustainability. Are there any plans or recommendations for ensuring the continued use of the ATP beyond the initial post-trial period? Discussing potential long-term strategies and considerations could provide a more comprehensive view of sustainment.

The discussion could be enhanced by elaborating on the policy implications of the findings. How can the insights gained from this study inform policy decisions at the clinic, regional, or national levels?

Reviewer #5: This is a fascinating qualitative study in the field of public health. It assesses the Sustainability of an evidence-based intervention supporting the transition to independent care for youth living with HIV in Kenya

The manuscript is well-written showing empirical evidence and meeting all standards of qualitative research.

However, the author should consider indicating the ethics approval number by the KNH/UoN in the methods section.

6. PLOS authors have the option to publish the peer review history of their article (what does this mean? ). If published, this will include your full peer review and any attached files.

**Do you want your identity to be public for this peer review?** For information about this choice, including consent withdrawal, please see our Privacy Policy .

Reviewer #1: **Yes: ** Adeniyi Adeniran MD, PhD

Reviewer #2: No

Reviewer #3: **Yes: ** Sumia Andleeb Abbasi

Reviewer #4: No

Reviewer #5: **Yes: ** FELIX GUMAAYIRI AABEBE

---

## [Decision Letter · Decision Letter 1]

3 Sep 2024

PGPH-D-24-00904R1

Sustainability of an evidence-based intervention supporting transition to independent care for youth living with HIV in Kenya

Dear Dr. Metje,

Thank you for submitting your manuscript to PLOS Global Public Health. After careful consideration, we feel that it has merit but does not fully meet PLOS Global Public Health’s publication criteria as it currently stands. Therefore, we invite you to submit a revised version of the manuscript that addresses the points raised during the review process.

We look forward to receiving your revised manuscript.

Kind regards,

Siyan Yi, MD, MHSc, PhD

Academic Editor

Additional Editor Comments (if provided):

Reviewers' comments:

Reviewer's Responses to Questions

**Comments to the Author**

1. If the authors have adequately addressed your comments raised in a previous round of review and you feel that this manuscript is now acceptable for publication, you may indicate that here to bypass the “Comments to the Author” section, enter your conflict of interest statement in the “Confidential to Editor” section, and submit your "Accept" recommendation.

Reviewer #1: All comments have been addressed

Reviewer #2: All comments have been addressed

Reviewer #3: All comments have been addressed

Reviewer #4: (No Response)

Reviewer #5: All comments have been addressed

2. Does this manuscript meet PLOS Global Public Health’s publication criteria ? Is the manuscript technically sound, and do the data support the conclusions? The manuscript must describe methodologically and ethically rigorous research with conclusions that are appropriately drawn based on the data presented.

Reviewer #1: Yes

Reviewer #2: Partly

Reviewer #3: Yes

Reviewer #4: Yes

Reviewer #5: Yes

3. Has the statistical analysis been performed appropriately and rigorously?

Reviewer #1: Yes

Reviewer #2: Yes

Reviewer #3: No

Reviewer #4: N/A

Reviewer #5: Yes

4. Have the authors made all data underlying the findings in their manuscript fully available (please refer to the Data Availability Statement at the start of the manuscript PDF file)?

Reviewer #1: Yes

Reviewer #2: Yes

Reviewer #3: No

Reviewer #4: Yes

Reviewer #5: Yes

5. Is the manuscript presented in an intelligible fashion and written in standard English?

Reviewer #1: Yes

Reviewer #2: Yes

Reviewer #3: (No Response)

Reviewer #4: Yes

Reviewer #5: Yes

6. Review Comments to the Author

Reviewer #1: The study background was presented logically and method was scientifically sound although, a qualitative study, the content and thematic analysis was sound.

Reviewer #2: Review Report

Title:Needs revision

Abstract:Lacks clarity

Background: Is relatively weak

Methods:Not explicit

Result and discussion: Lacks logical flow, brief and concise presentation

Regards,

Reviewer #3: - Considerbreaking down long sentences into shorter, more direct ones can enhance clarity.

- Improve the transition between the background on YLH challenges and the introduction of the ATTACH study. A brief connecting sentence could help maintain the flow of the narrative.

- A brief statement on how the findings of this study might influence future interventions or public health strategies could add depth to the introduction.

- Addressing the challenges of high staff turnover by ensuring ongoing training and support for new staff is essential for the long-term success of ATP implementation.

- Integrating ATP tools into existing electronic medical systems and aligning them with clinic workflows will improve documentation consistency and overall program fidelity.

-The discussion could benefit from deeper exploration of the specific challenges faced, particularly regarding the integration of ATP into existing EMR systems. More details on how these integration issues specifically impacted the ATP’s use and sustainment would be valuable.

-While comparisons with other studies are made, the discussion could be strengthened by providing more detailed comparisons with similar programs or interventions in different settings.

- There could be a more detailed discussion on how the findings align or contrast with specific studies on similar interventions. This would provide a more nuanced understanding of where the ATP stands in relation to other EBIs.

- Recommendations for addressing the specific barriers identified (e.g., workforce shortages, high staff turnover, EMR integration) could be more detailed. Providing concrete strategies or examples from other programs that have successfully overcome these barriers would enhance the practical utility of the discussion.

- The results section mentions that 30 HCWs participated in the study, but it does not address the justification for this sample size. Providing information on how this number was determined (e.g., power analysis, feasibility) would strengthen the validity of the findings.

- While gender and cadre distributions are provided, there is no discussion on how these distributions might influence the results. For instance, how might the predominance of female HCWs (73.3%) and the various cadres (e.g., peer mentors vs. clinical officers) impact the ATP’s implementation and feedback?

- Although the training sources are mentioned, there is no further analysis of how the training by ATTACH study staff versus other HCWs might have impacted the implementation and effectiveness of the ATP. This could be an important factor in understanding variability in responses.

Reviewer #4: Sustainability of an evidence-based intervention supporting transition to independent care for youth living with HIV in Kenya

Thank you for the manuscript. This is a very important study for the region. The manuscript is generally well-written, but I have a few observations which, when addressed, will improve the manuscript. Most are minor issues to be explained or addressed.

Introduction

While the background is focused on Kenya, it could benefit from a broader perspective by briefly discussing how similar challenges and interventions are being addressed in other regions or countries. This would position the study within a global context and increase its relevance.

While the background mentions the lack of evidence on EBI sustainment post-trial, it would benefit from a brief discussion of why sustainment is particularly challenging in low-resource settings like Kenya.

The link between the challenges faced by YLH and the need for sustainment of EBIs could be made more explicit earlier in the section. This would help the reader immediately understand the importance of the study.

The section on the challenges associated with transitioning from pediatric to adult care is well-explained, but it could be enhanced by providing more specific examples of how these challenges manifest in practice. For instance, explaining how anxiety and reluctance to transition contribute to treatment interruptions could add depth to the discussion.

The background introduces the ATTACH study as a hybrid implementation-effectiveness cluster randomized clinical trial (cRCT). However, it would be helpful to briefly explain what a hybrid implementation-effectiveness trial is, as not all readers may be familiar with this study design. This could be done in a sentence or two.

Methods

While purposive sampling is appropriate for qualitative studies, it's essential to clearly articulate the specific criteria used for selecting participants. How were participants selected based on their roles, experience with ATP, and satisfaction scores?

Consider providing a justification for the sample size of two to three HCWs per clinic. Was this based on prior research, power calculations, or theoretical saturation?

While the authors mention that the interview guide was informed by CFIR, it would be beneficial to provide more details about its content. What were the key themes explored in the interviews?

The authors mention using English, Kiswahili, and Dholuo. Were there specific criteria for language selection? How were potential language barriers addressed during the interviews and transcription process?

The description of interviewer training is brief. What specific training did the interviewers receive in qualitative interviewing techniques, CFIR, and cultural sensitivity?

The authors mention a hybrid approach to codebook development. It would be helpful to elaborate on how the deductive and inductive coding processes were integrated. Were there specific guidelines for when to use each approach?

The methods mention that some CFIR constructs were removed from the codebook. It would be helpful to specify which constructs were removed and why, as this could impact the interpretation of the results.

While the authors mention regular meetings to discuss codes and resolve discrepancies, it's important to report specific measures of inter-rater reliability (e.g., Cohen's kappa) to assess the consistency of coding.

The description of reflexivity is brief. It would be beneficial to provide more details about how researchers addressed their personal biases and how these biases might have influenced the interpretation of data.

While the authors mention ethical approval and informed consent, it's important to address potential power dynamics between researchers and participants, particularly in a hierarchical setting like healthcare.

The authors mention plans for a collaborative stakeholder workshop. It would be helpful to elaborate on how findings from this qualitative study will be shared with participants and other stakeholders to inform future interventions.

While there is a reference to inclusivity in global research, a summary or key points from the Supporting Information would provide context on how inclusivity was addressed in the study.

Reviewer #5: The manuscript has no dual publication, research, or publication ethics issues. Also, my earlier review issue about the ethical approval number has been addressed.

7. PLOS authors have the option to publish the peer review history of their article (what does this mean? ). If published, this will include your full peer review and any attached files.

**Do you want your identity to be public for this peer review?** For information about this choice, including consent withdrawal, please see our Privacy Policy .

Reviewer #1: **Yes: ** Adeniyi Adeniran Phd.

Reviewer #2: No

Reviewer #3: No

Reviewer #4: No

Reviewer #5: No

---

## [Decision Letter · Decision Letter 2]

5 Dec 2024

Sustainability of an evidence-based intervention supporting transition to independent care for youth living with HIV in Kenya

PGPH-D-24-00904R2

Dear Alina Metje,

We are pleased to inform you that your manuscript 'Sustainability of an evidence-based intervention supporting transition to independent care for youth living with HIV in Kenya' has been provisionally accepted for publication in PLOS Global Public Health.

Best regards,

Henry Zakumumpa, PhD

Academic Editor

Reviewer Comments (if any, and for reference):

Reviewer's Responses to Questions

**Comments to the Author**

1. If the authors have adequately addressed your comments raised in a previous round of review and you feel that this manuscript is now acceptable for publication, you may indicate that here to bypass the “Comments to the Author” section, enter your conflict of interest statement in the “Confidential to Editor” section, and submit your "Accept" recommendation.

Reviewer #4: All comments have been addressed

2. Does this manuscript meet PLOS Global Public Health’s publication criteria ? Is the manuscript technically sound, and do the data support the conclusions? The manuscript must describe methodologically and ethically rigorous research with conclusions that are appropriately drawn based on the data presented.

Reviewer #4: Yes

3. Has the statistical analysis been performed appropriately and rigorously?

Reviewer #4: Yes

4. Have the authors made all data underlying the findings in their manuscript fully available (please refer to the Data Availability Statement at the start of the manuscript PDF file)?

Reviewer #4: Yes

5. Is the manuscript presented in an intelligible fashion and written in standard English?

Reviewer #4: Yes

6. Review Comments to the Author

Reviewer #4: The author addressed all corrections.

7. PLOS authors have the option to publish the peer review history of their article (what does this mean? ). If published, this will include your full peer review and any attached files.

**Do you want your identity to be public for this peer review?** For information about this choice, including consent withdrawal, please see our Privacy Policy .

Reviewer #4: **Yes: ** KHADIJAT ADELEYE
